# Could the Visual Differential Attention Be a Referential Gesture? A Study on Horses (*Equus caballus*) on the Impossible Task Paradigm

**DOI:** 10.3390/ani8070120

**Published:** 2018-07-17

**Authors:** Alessandra Alterisio, Paolo Baragli, Massimo Aria, Biagio D’Aniello, Anna Scandurra

**Affiliations:** 1Department of Biology, University of Naples Federico II, 80126 Naples, Italy; alessandra.alterisio@unina.it (A.A.); annascan@hotmail.it (A.S.); 2Department of Veterinary Sciences, University of Pisa, 56124 Pisa, Italy; paolo.baragli@unipi.it; 3Department of Economics and Statistics, University of Naples Federico II, 80126 Naples, Italy; aria@unina.it

**Keywords:** impossible task paradigm, horse–human relationship, communication, horse, helping request, attentional state

## Abstract

**Simple Summary:**

We tested horses on the impossible task paradigm, a drawback affecting the decision-making process in animals. We used the direction of the horse’s ear cup as an indicator of its visual attention in terms of visual selective attention when both ears were directed at the same target and the visual differential attention when the ears were directed differentially to the persons and to the experimental tools. We aimed to evaluate whether the latter behavior could be considered a possible support to solve the problem. The visual differential attention was the most frequent behavior when the resource was unreachable, which supports the view that this gesture could be linked to a request for help from humans to find the solution to the task. Our procedure proved to be a useful way to understand how horses try to attract human attention when they are in a restricted environment, a typical situation for horses living in stables.

**Abstract:**

In order to explore the decision-making processes of horses, we designed an impossible task paradigm aimed at causing an expectancy violation in horses. Our goals were to verify whether this paradigm is effective in horses by analyzing their motivation in trying to solve the task and the mode of the potential helping request in such a context. In the first experiment, 30 horses were subjected to three consecutive conditions: no food condition where two persons were positioned at either side of a table in front of the stall, solvable condition when a researcher placed a reachable reward on the table, and the impossible condition when the food was placed farther away and was unreachable by the horse. Eighteen horses were used in the second experiment with similar solvable and impossible conditions but in the absence of people. We measured the direction of the horse’s ear cup as an indicator of its visual attention in terms of visual selective attention (VSA) when both ears were directed at the same target and the visual differential attention (VDA) when the ears were directed differentially to the persons and to the table. We also included tactile interaction toward table and people, the olfactory exploration of the table, and the frustration behaviors in the ethogram. In the first experiment, the VDA was the most frequent behavior following the expectancy violation. In the second experiment, horses showed the VDA behavior mostly when people and the unreachable resource were present at the same time. We speculate that the VDA could be a referential gesture aimed to link the solution of the task to the people, as a request for help.

## 1. Introduction

Several studies have highlighted the cognitive abilities of horses in social interactions with humans [1]. However, only a few studies have focused on interspecific referential communication between horses and humans. One of the ways to analyze referential communication between animals and humans is the impossible task (IT) paradigm, which entails an unsolvable task preceded by a number of trials in which the subject learns how to obtain a reward independently without previous training. The IT paradigm is a useful tool in comparative studies of cognitive abilities of animals vis-a-vis expectancy violation, a drawback affecting the decision-making process in animals, wherein they have to choose whether to try solving the previously learned task autonomously or to ask for support from a potential human helper [2]. The expectancy violation was not elicited (or at least not verified) in previous studies dealing with unsolvable tasks in horses [3,4], since the horses were not experienced before with a solvable phase in which they learn to solve a simple problem (see also [5,6]). To the best of our knowledge, horses have never been subjected to an IT paradigm as described above [2], which we have performed in this study to explore their decision-making processes.

Malavasi and Huber (2016) [3] showed that horses use referential gestures to manipulate the attention of a human recipient in an attempt to obtain an unreachable resource, by increasing the rate of gaze alternation between the bucket containing food and the experimenter. The studies of both Malavasi and Huber (2016) [3] and Ringhofer and Yamamoto (2017) [4] considered the position of the head axis to determine the binocular gazing of horses as a behavioral measure of their visual communication with humans. However, since predicting the direction of the horse’s gaze is not easy, the ear orientation was used to determine the direction of its visual attention [7]. Therefore, in previous studies, the gazing behavior was scored on the basis of head direction by the orientation of both [4] or at least one of the ears [3] toward the target. We have introduced the possibility that the horse’s visual attention may also be inferred by the ears’ differential positioning indicating simultaneous focusing on two different targets, without the need of gaze alternation (i.e., requiring eye and/or head movement). This is similar to the concept of “divided attention”, wherein horses concentrate on different stimuli simultaneously [8]. We have considered the visual differential attention (VDA) in addition to visual selective attention (VSA), in which the horse’s attention is centered on a single target, as indicated by the unidirectional orientation of the head and both ears [9].

Our goal was to explore the decision-making processes in the IT aimed at causing an expectancy violation in horses. To this aim, we first verified whether the IT paradigm was effective in horses by analyzing their motivation in trying to solve the task, as well as any change in their behavior when the task became impossible. We then evaluated whether and how the horses related to humans when trying to solve the task in the impossible condition.

## 2. Materials and Methods

In the first experiment, 30 horses (16 females, 12 geldings, and 2 males of different breeds; mean age = 12.1 ± 0.9 years) provided by three different riding schools (20, 3, and 7 subjects) were used and were deprived of food for at least 4 h prior to the task. A table (85 × 85 cm) was placed outside the stall, close to the entrance, and the horse was prevented from exiting its stable by a rope. Before the test, the subjects were acclimatized for about five minutes to the table, on which the researcher placed food (a carrot or an apple) that the horse could reach by stretching its neck. This helped determine the horse’s motivation toward the food and allowed the animal to familiarize itself with the experimental setup.

The horses were subjected to three consecutive conditions to elicit an expectancy violation in the last. In the no food condition (NF) lasting 60 s, the caretakers (three different males from three different riding centers) and a female stranger were positioned at either side of the table. They stood motionless facing each other in a neutral position and ignoring the horse as best as they could (Figure 1).

In the following solvable condition (SC), a researcher (unknown to the horses) approached the experimental area and displayed the reward (food) to the subject by placing it on the table, such that the horse could reach it independently. This sequence was repeated thrice, and the duration of this phase was variable according to the time taken by the horse to eat the food. The impossible condition (IC) was the same as the previous, but the researcher placed the food away from the reach of the horse. This phase lasted 60 s and aimed to evaluate the horse’s behavior in an expectancy violation. The researcher approached and moved away from the experimental area randomly from the right or the left, taking care to disappear in the easiest and fastest way from the horse’s view. All conditions were in the same order and were consecutive.

To evaluate the effect of the people in the test, we performed an experiment 2 including an additional solvable condition 2 (SC2) and impossible condition 2 (IC2), with no humans but 18 more horses recruited from two different stables (2 females, 14 geldings, and 2 males of different breeds; mean age = 15.2 ± 1.1 years), provided by two different riding schools (5 and 13 subjects). In this case, the code for the VDA would be that horses have one ear on the table and one ear turned to the place where the humans had previously stood. The solvable conditions (i.e., SC and SC2) were only aimed to trigger the expectancy violation in the impossible conditions and were not standardized nor included in the analysis.

All applicable international, national, and/or institutional guidelines for the care and use of animals were followed. This study has been approved by the Ethical Animal Care and Use Committee of the University of Pisa (protocol number 63714/2016).

The whole tests were recorded using a camera (Sony Handycam HDR-CX115, Tokyo, Japan) located at a distance of 4 m in front of the horse. The videos were analyzed by a trained researcher, and the horse’s behaviors (Table 1) were coded using Solomon Coder^®^ beta 16.06.26 (ELTE TTK, Budapest, Hungary). Since the SC was only aimed to correctly trigger and study the expectancy violation in the IC, it was not temporally standardized and was not included in the analysis. A second independent blind coder randomly analyzed 25% of the samples, and a very high agreement on response emerged from the inter-observer reliability tests for all behavioral parameters (from 96% to 99%).

A statistical analysis was conducted on comparing the behaviors directed to the stranger and the caretaker in the IC by using a pairwise Wilcoxon test, since the Shapiro-Wilk test revealed that the data were not normally distributed. To record statistical differences following the expectancy violation, all behavioral parameters were compared in a between condition approach using the Wilcoxon two-sample paired test to determine differences between the NF and IC. A within-condition analysis to explore the decision-making processes and the communication methods during the expectancy violation in the IC was performed using Friedman’s test, with a Wilcoxon post-hoc test (Bonferroni corrected) comparing the horse’s behaviors involving people. The SC was mandatory to correctly trigger and study the expectancy violation in the following phase and was not considered in the analysis.

## 3. Results and Discussion

The pairwise Wilcoxon test recorded no significant difference in the behavior of the horses toward the caretakers and the stranger; therefore, all behaviors toward humans were considered as a whole. The Wilcoxon test showed no significant changes in olfactory exploration following the expectancy violation (i.e., from the NF to the IC). However, we observed a clear increase in tactile interaction (frequency: *W* = 271, *z* = 3.46, *p* < 0.001; duration: *W* = 286, *z* = 2.81, *p =* 0.005) and VSA (frequency *W* = 382, *z* = 4.07, *p <* 0.001; duration *W* = 380, *z* = 3.50, *p <* 0.001). Furthermore, the horses also showed such behaviors toward the table sooner (latency tactile interaction: *W* = 308, *z* = 3.35, *p =* 0.001; latency VSA: *W* = 394, *z* = 4.34, *p <* 0.001). The horses’ VDA behavior involving the people and the table showed the same trend, notably increasing from the NF to IC (frequency: *W* = 406, *z* = 3.57, *p <* 0.001; duration: *W* = 404, *z* = 4.03, *p <* 0.001), while decreasing in latency (*W* = 356, *z* = 2.54, *p =* 0.011). The frustration behaviors significantly increased from the NF to the IC (frequency: *W* = 334, *z* = 2.99; *p =* 0.003; duration: *W* = 362, *z* = 2.65, *p =* 0.008), and occurred sooner (latency: *W* = 345, *z* = 2.31, *p =* 0.021). Altogether, an increase in the frustration behaviors, together with the behaviors involving the table from the NF to the IC, indicates an attempt of the horses to find the solution in the new unexpected situation, supporting that the expectancy violation was effectively triggered in the horses.

A within-condition analysis to explore the decision-making processes and the communication methods during the expectancy violation in the IC revealed that the tactile interaction, VSA, and VDA (i.e., to people-table) were expressed in a significantly different manner (frequency: *χ*^2^ = 53.32, *df* = 2, *p <* 0.001; duration: *χ*^2^ = 33.80, *df* = 2, *p <* 0.001; latency: *χ*^2^ = 32.47, *df* = 2, *p <* 0.001). Post-hoc tests showed that the horses used VSA more than tactile interaction (frequency: *W* = 371.5, *p <* 0.001; duration: *W* = 321.5, *p =* 0.021), while VDA was more expressed than either VSA (frequency: *W* = 461.5, *p <* 0.001; duration: *W* = 439, *p <* 0.001) or tactile interaction (frequency: *W* = 465, *p <* 0.001; duration: *W* = 462, *p <* 0.001). Finally, VSA toward people happened sooner than tactile interaction (*W* = 369, *p*
*<* 0.001), whereas the horses’ VDA had a shorter latency than both VSA (*W* = 407, *p <* 0.001) and tactile interaction (*W* = 456, *p <* 0.001). Overall, the results show that the VDA was the behavior that happened most frequently, which could indicate that it was a referential gesture toward humans, but it could not be related to the people in the test.

In order to exclude the second hypothesis, we performed a supplementary experiment including additional solvable condition 2 (SC2) and impossible condition 2 (IC2), with no humans. The Wilcoxon test again showed no differences between the human subjects involved in the test in NF and IC, which allowed the data to be grouped. The Friedman test revealed that the VDA involving the table differed significantly in the three conditions (NF, IC, and IC2) in frequency (*χ*^2^ = 16.36, *df* = 2, *p <* 0.001) and duration (*χ*^2^ = 18.11, *df* = 2, *p <* 0.001) but not in latency (*χ*^2^ = 0.52, *df* = 2, *p =* 0.779). Post-hoc testing showed that horses used the VDA more in the IC rather than the NF (frequency: *W* = 168, *p <* 0.001; duration: *W* = 171, *p <* 0.001) and IC2 (frequency: W = 158, *p =* 0.002; duration: *W* = 152, *p =* 0.006) (Figure 2). Therefore, the horses showed the VDA behavior mostly when people and the unreachable resource were present at the same time.

It is well known that dogs use gazing as a request for help in the IT paradigm [5,6,10]. Horses also show increased interest toward the experimenter when experiencing difficulties in a problem-solving task by focusing their gaze on the human’s face [11], implying that horses communicate their request for help to humans visually [12]. This suggests that VDA could be another way that horses use to communicate with humans to request help, presenting the possibility for the first time of this behavior as a referential gesture toward humans.

The VDA could be a specific behavior that horses use to communicate in restricted environments, since their gaze alternation has been observed in experimental settings in larger areas [3,4]. Comparing our results with those of previous studies [3,4], we cannot exclude that communication modalities (i.e., gaze alternation, VSA, VDA, and tactile interaction) could be context-dependent, based on the physical distance between sender and recipient. In our experimental setting, gaze alternation was hardly observed and was not included in the ethogram. However, an alternative explanation why horses mostly used VDA in the IC could be that they paid closer attention to the humans because they were extending their neck to reach the food, which is a more vulnerable position and results in increased vigilance. Alternatively, they looked at the humans to make sure they were not going to steal the food.

## 4. Conclusions

The visual differential attention was the most frequent behavior when the resource was unreachable, which supports the view that such a gesture could be a way to link the solution of the task to the people, as a request for help. However, more studies should be done before considering definitively the visual differential attention as a referential gesture. In any case, our procedure proved to be a useful way to understand how horses try to attract human attention when they are in a restricted environment, a typical situation for horses living in domestic environments.

## Figures and Tables

**Figure 1 animals-08-00120-f001:**
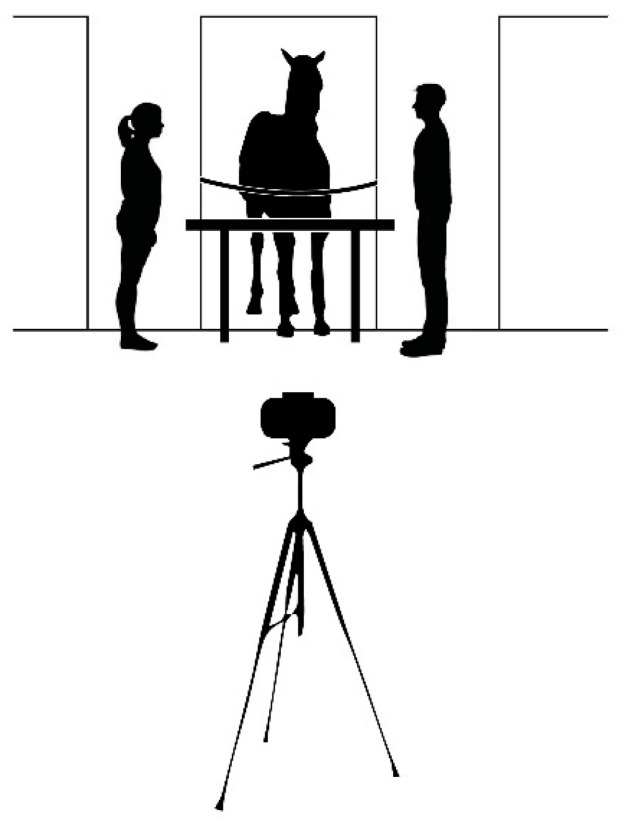
The experimental setting of the no food condition (NF), solvable (SC), and impossible (IC) conditions. The apparatus consisted of a table (85 × 85 cm), placed close to the entrance of the stall, which was freely accessible by horses. The caretaker was positioned to the left of the horse, while the unknown person was positioned on the right. A rope was placed across the stable door. The camera was located at a distance of 4 m in front of the horse.

**Figure 2 animals-08-00120-f002:**
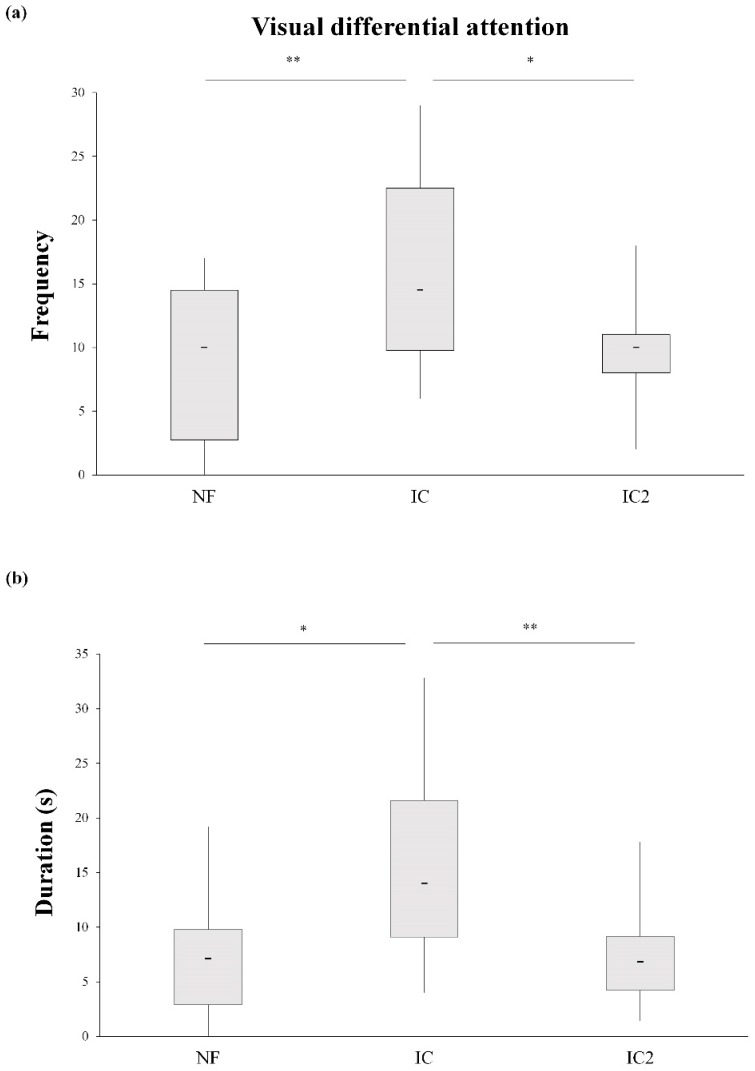
Comparison of the visual differential attention in the no food condition (NF), the impossible condition with humans (IC), and the impossible condition with no humans (IC2) (experiment 2) in frequency (**a**) and duration (**b**). The asterisks indicate statistically significant differences according to repeated-measures Friedman test with Wilcoxon pairwise (Bonferroni corrected) as the post-hoc test. Bold horizontal lines: medians; grey boxes: quartiles; thin vertical lines: minimum and maximum values. * *p* < 0.05; ** *p* < 0.001.

**Table 1 animals-08-00120-t001:** Behaviors recorded during the test with descriptors.

Behaviors	Description
Olfactory exploration	The horse sniffed with one or both nostrils the table at a distance of about 10 cm or less with the mouth closed.
Tactile interaction	The horse touched the table or the people (the caretaker and the stranger) with its muzzle and its mouth by licking, biting, nibbling, and moving lips.
Selective attention	Both ear cups of the horse pointed toward the same target—table, people.
Differential attention	The two ears were directed in different directions—table-people.
Frustration behaviors	Head nods and head shakes, vacuum chewing, pawing, snorting, and putting ears back.

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
