# Peer review of "Could the Visual Differential Attention Be a Referential Gesture? A Study on Horses (Equus caballus) on the Impossible Task Paradigm"

_animals, 2018, doi:10.3390/ani8070120_

Round 1

Reviewer 1 Report

This is an interesting research article focusing on the cognitive ability of horses to interact with humans in the request for help in solving a task. This is novel research that investigates an aspect of horse-human interaction that has not been undertaken. while the results are preliminary, they lend a basis for further research in this area. However the article could be improved by clearer explanations in all parts of the manuscript, and attention to the limitations or other explanations of the results.

Simple summary

L13 – could this be explained in simpler terms?

L17 – the term apparatus is confusing. Can a different term be used that explains it better, or else explain clearly in the text what this is referring to. Change throughout the manuscript

L17 – delete the word “were”  We aimed to evaluate…

L18 – reword “…as a referential gesture toward humans as a request for help.”

L19-20  awkward english. perhaps reword as "which supports the view that this gesture could be linked to a request for help from humans to find the solution to the task."

L23 – delete this line. Period after stables.

Abstract

L24 – add “In the first experiment, 30 horses…”

L28 – add “18 horses were used in the second experiment…”

L29 – would the word similar be appropriate rather than additional?

L29 – replace "used" with "measured"

L32 – replace "used" with "measured"

Intro

L43 – would be useful to refer to these studies here, even if briefly

L49-50 - outline the context of these past experiments and how you infer that the expectancy violation was not elicited.

L54 – you want to outline change in behaviour in exp 1, but this sounds like the same objective as your exp 2. Maybe remove the words “toward humans” here

L56 – include reference [3]

L58 – would Malavasi and Huber’s research not qualify as an IT? But on line 51 you say that it hasn’t been done

L59 – include reference [3]

L60 – include reference [4]

L60 – remove comma after horse “…binocular gazing of horse as a direction…”

M&M

L73 – need details of the horses – age, gender, breed

L78 – add the word to “…allowed the animal to familiarize…”

L81 – how many horses were tested at each of the riding centres?

L81 – did the stranger humans alternate which side of the table they were standing on?

Fig 1 - please add more information to the caption to make it understandable even without having to read the text. ie. include information about horses, humans, table, etc.

L87 – was the researcher known to the horse?

L90 – was the SC always followed by the IC?  was the order always the same? when was NF recorded? what was the time lapse between application of treatments?

L96 – were the videos analyzed by a single researcher?

L96 – replace “their” with “the horses’”

Table 1 –

olfactory exploration - how could the distance of 10cm be determined if the camera was directly in front of the horse at a distance of 4m?

tactile exploration – what were the people instructed to do if/when the horse touched them?

Differential attention – what if the ears were not pointed at either the apparatus or the people?

Statistical tests used to analyze the data should be included in the M&M section, not in the results. Also, did you account for location in the analyses, since you mention three different stables?

Results

L108 - give statistical value of outcome of univariate test for normality (or lack of it)

L110 – is this observed increase in tactile interaction from NF to IC? Please state.

L112 – clarify that the horses showed behaviours toward the apparatus sooner in the IC than the NF (I think this is what you mean)

L119-121 - this sentence belongs in the discussion

L132-135 - again, this should appear in the discussion.

L133 – I don't think you can surmise that VDA is the preferred behaviour - only that it was the behaviour that happened most frequently.

L134 – the phrase “…a general increase in movement…” is vague. What do you mean by this?

L136-141 - this should be in the methodology with a much better description to understand what exactly you did. perhaps headings (eg. exp 1, exp 2, exp 3) would be useful

L137 – add the word “an additional solvable….”

L138 – provide information on the horses (age, breed, gender)

L141 – reword “…were not standardized nor included in the analysis”

L142 – your sentence above states that no humans were involved – so how would no differences be calculated?

L146 – could the fact that the horses used VDA more in IC than NF or IC2 be because they were becoming accustomed to the experimental situation and realizing that the task was insolvable, hence not trying to figure it out anymore. You could test this by alternating the order of presentation of IC and IC2 (if i'm understanding that IC was with people and IC2 was without).

L148-149 - move to discussion

Fig 2 - need a reference to this figure within the text. Include more detail in this caption as per above. Also you have not referred to exp 2 anywhere else. see comment above re dividing methodology and results into sub-headings

L168 - your methodology does not suggest that you collected data on gaze alternation

L175 – add s to supports and gestures

L176 – reword “…more studies should be done before considering…”

L179-180 – reword “…living in the domestic environment.”

Author Response

Attachment.

Reviewer 2 Report

Dear Authors,

the paper “Could the visual differential attention be a referential gesture? A study on horses (Equus caballus) in the impossible task paradigm” presents the preliminary results of a research aiming to identify communicative methods of horses to ask help from humans to solve an impossible task.

The paper is of interest for the journal readers and the experiment is well conducted. However, the manuscript need some improvements in order to make it clearer.

Abstract: the abstract is not clear enough nor attractive for the reader. It is not self standing, so I suggest to largely rearrange it. In particular, the aim of the study is not clearly stated. The second part of the study is mentioned but no results from this are presented; the sentence (L28-29) could be removed or, either, a sentence describing the results from this second experiment should be added.

Introduction: the aim of the study should be more clearly specified in the introduction.

L53-56: I suggest moving these contents at the end of the introduction, rearranging the remaining part of the paragraph consequently.

L56: the reference should be reported as [3].

L59-60: the references should be reported as [3, 4].

Materials and methods: this section should give the reader more information about the experiment. In particular, I suggest to add:

·         information about the horses involved (sex, age, breed…);

·         information about the horses housing and working routine;

·         detailed description of statistical analysis.

L96: I suggest to replace “their” with “horses”

Result and discussion: rearrange this section as needed after including the statistical analysis details in Material and methods section.

L136-149: this second experiment should be described in the Material and methods section. This experiment could be presented adding other Material and methods and Result and discussion sections after the results of the first one and then a general discussion could be added, such as in Ringhofer & Yamamoto (2017). Alternatively, only one Material and methods section could be maintained but the description of the second experiment needs to be added here. The discussion of the results should be rearranged consequently.

L168-172: this paragraph have the intention to present alternative explanations for the results; however it is not well linked with the rest of the discussion and alternative explanations are not in-depth analyzed (i.e. references?). I would suggest to expand this paragraph a little in order to make it clearer and, if available, add some references.

Figure 2. The quality of the picture is not enough: text is hard to read, lines of the graph are difficult to distinguish.

References:

Modify the reference 4. Ringhofer, M.; Yamamoto, S. Domestic horses send signals to humans when they face with an unsolvable task. Anim Cogn 2017, 20, 397-405, DOI: http://dx.doi.org/10.1007/s10071-017-1074-x

In the original publication, the article title was incorrectly published as ‘Domestic horses send signals to humans when they face with an unsolvable task’. The correct title should read as ‘Domestic horses send signals to humans when they are faced with an unsolvable task’.

Author Response

Attachment

Round 2

Reviewer 2 Report

Dear Authors,

thank you for your revision. I believe that the manuscript has improved and is now clearer.

Best wishes